# A Trunk Detection Method for *Camellia oleifera* Fruit Harvesting Robot Based on Improved YOLOv7

Yang Liu [1,2], Haorui Wang [1], Yinhui Liu [3], Yuanyin Luo [1], Haiying Li [1], Haifei Chen [1], Kai Liao [1] and Lijun Li [1,*]

1  School of Mechanical and Electrical Engineering, Central South University of Forestry and Technology, Changsha 410004, China; yangliu@csuft.edu.cn (Y.L.)
2  Hunan Automotive Engineering Vocational College, Zhuzhou 412001, China
3  Zhongqing Changtai (Changsha) Intelligent Technology Co., Ltd., Changsha 410116, China
*  Correspondence: junlili1122@163.com; Tel.: +86-13974841598

**Abstract:** Trunk recognition is a critical technology for *Camellia oleifera* fruit harvesting robots, as it enables accurate and efficient detection and localization of vibration or picking points in unstructured natural environments. Traditional trunk detection methods heavily rely on the visual judgment of robot operators, resulting in significant errors and incorrect vibration point identification. In this paper, we propose a new method based on an improved YOLOv7 network for *Camellia oleifera* trunk detection. Firstly, we integrate an attention mechanism into the backbone and head layers of YOLOv7, enhancing feature extraction for trunks and enabling the network to focus on relevant target objects. Secondly, we design a weighted confidence loss function based on Facol-EIoU to replace the original loss function in the improved YOLOv7 network. This modification aims to enhance the detection performance specifically for *Camellia oleifera* trunks. Finally, trunk detection experiments and comparative analyses were conducted with YOLOv3, YOLOv4, YOLOv5, YOLOv7 and improved YOLOv7 models. The experimental results demonstrate that our proposed method achieves an mAP of 89.2% , Recall Rate of 0.94, F1 score of 0.87 and Average Detection Speed of 0.018s/pic that surpass those of YOLOv3, YOLOv4, YOLOv5 and YOLOv7 models. The improved YOLOv7 model exhibits excellent trunk detection accuracy, enabling *Camellia oleifera* fruit harvesting robots to effectively detect trunks in unstructured orchards.

**Keywords:** trunk detection; *Camellia oleifera*; attention mechanism; CBAM; Facol-EIoU; improved YOLOv7

## 1. Introduction

*Camellia oleifera* an important oil-bearing crop, is cultivated extensively in the south of China [1]. The morphology of *Camellia oleifera* is characterized by evergreen shrubs or medium-sized trees and its leaves are elliptical, oblong, or inversely ovate in shape [2]. The harvesting of *Camellia oleifera* is the most expensive stage, demanding significant human and financial resources. The harvesting of *Camellia oleifera* fruit is particularly challenging and hazardous due to the complex growth environments, which are often found in hilly and mountainous areas [3]. Consequently, there is a pressing need to develop a mechanized and intelligent picking method that can reduce labor requirements and enhance harvesting efficiency for *Camellia oleifera*. The development of the *Camellia oleifera* fruit harvesting industry has witnessed the design and application of numerous mechanized fruit-picking machines [4]. In our previous work, we demonstrated that vibration-based methods are the most effective approach for mechanized harvesting of *Camellia oleifera*, leading us to design a *Camellia oleifera* fruit harvesting robot that utilizes vibration to pick fruits, thereby achieving mechanized harvesting [5]. However, the current robot still requires manual intervention to select the vibration points on the trunk, which limits its level of intelligence. Therefore, the rapid and accurate detection of *Camellia oleifera* trunks using computer vision

methods has emerged as a crucial technology for harvesting robots and has become the primary focus of research.

Exiting methods for trunk and crop detection mainly rely on imaging process technology [6–8]. Several methods have been proposed and improved for detecting various crops, including *pineapple* [9], *grape* [10,11], *apple* [12–15], *kiwifruit* [16,17] and others. While there has been some research on the detection of *Camellia oleifera* fruit and flowers [3,8], there is limited study on the detection of *Camellia oleifera* trunks. In order to assist the *Camellia oleifera* harvesting robot in locating the vibration point, trunk detection is the crucial initial step. There are two main learning methods of crop detection, traditional machine image processing and deep learning image processing, which aim to identify the class and determine the position of targets in the image [18]. Traditional methods, such as support-vector-machine-based (SVM) [19], histogram of oriented gradient (HOG) [20], deformable parts model (DPM) [21] and others, have commonly been employed for image detection and segmentation. These methods can mitigate the impact of illumination changes on image detection. However, traditional detection methods suffer from the drawback of being time-consuming.

Deep learning has gained widespread application in the field of computer vision, particularly in target detection, where it has achieved remarkable success. Convolutional neural networks (CNN) have been applied to detect various fruits in images. Models such as AlexNet [22], VGGNet [23] and Inception [24] have been updated to enhance recognition accuracy for harvesting robots in field experiments. With advancements in learning algorithms, learning-based object detection methods can be categorized into two types: the one-stage method and two-stage method. R-CNN, Fast R-CNN and Faster R-CNN [25] use the two-stage method. While these methods improve recognition accuracy, they suffer from slow detection speed, making them unsuitable for real-time applications. SSD [26] and the YOLO series [27] are the typical one-stage object-detection methods, which divide the image into regions and promptly determine the object boundaries and classification probabilities of for each object. They offer faster detection speeds but may sacrifice a slight decrease in recognition accuracy compared to two-stage methods.

In terms of balancing detection accuracy and recognition speed, one-stage object detection algorithms are more suitable for vision systems in harvesting robots. The YOLO serial algorithms propose the use of an end-to-end neural network that simultaneously predicts bounding boxes and class probabilities. Wu et al. [8] applied the YOLOv7 model to recognize and locate the *Camellia oleifera* fruit and improve the detection accuracy. YOLOv7 [28] is the latest detector of the YOLO series, designed with a trainable bag-of-freebies. This design enables real-time detectors to significantly enhance accuracy without increasing the inference cost. It incorporates extended compound scaling techniques to effectively reduce the number of parameters and calculations in the target detector, thereby greatly improving detection speed. Therefore, we have applied YOLOv7 to detect *Camellia oleifera* trunks in our work.

The primary motivation of this study is to develop a robust and reliable approach to detect *Camellia oleifera* trunks using an improved YOLOv7 model. This improvement aims to enhance trunk detection performance in unstructured environments. The contributions of this study are summarized as follows:

(1) The creation of a dataset comprising manually annotated visible images of *Camellia oleifera* trunks and fruits captured in *Camellia oleifera* orchards.

(2) An algorithm for trunk detection was proposed based on the improved YOLOv7 model using monocular vision images. This method enables *Camellia oleifera* harvesting robots to identify and detect trunks. An attention mechanism, specifically, a CBAM (Convolutional block attention module) module, is incorporated in the backbone of YOLOv7 to enhance the detection accuracy of *Camellia oleifera* trunks.

(3) The application of the Facol-EIoU loss function to replace the loss function in the improved YOLOv7 network, further enhancing the detection of *Camellia oleifera* trunks. A comparison is made between the Precision (P), Recall(R), F1 and Detection Speed of

the improved YOLOv7 and other algorithms, including YOLOv3, YOLOv4, YOLOv5 and YOLOv7.

## 2. Materials and Methods

### 2.1. Camellia oleifera Trunk Image Acquisition

The images of *Camellia oleifera* trunks used in this study were obtained from the *Camellia oleifera* base located in Zhentou Village, Liuyang county, Changsha city, Hunan Province. The *Camellia oleifera* in the base were planted at an approximate distance of three meters from each other, which creates a suitable environment for the robot to perform harvesting tasks. All pictures were captured using a monocular camera (Micovision Co., Ltd., Washington, DC, USA) with a resolution of 4096 × 3072 pixels. The camera was mounted over the *Camellia oleifera* trunks at a fixed distance of 130 cm and a height of 100 cm, as shown in Figure 1, in order to capture clear trunk data. A total of 1500 pictures of *Camellia oleifera* trunks were collected under different conditions, including single trunks with front-light, single trunks with back-light, clusters of trunks with front-light and clusters of trunks with back-light, as shown in Figure 2. The resolution of the trunk images was reduced to 640 × 640 pixels to meet the requirements of the YOLOv7 model.

### 2.2. Data Annotation

The *Camellia oleifera* trunks in the images were individually annotated as a single class. Data labeling involves the process of adding tags or labels to raw data. In this study, the datasets were labeled using LabelImage tool and followed the Pascal Visual Object Classes format [29]. The annotation samples of labeled *Camellia oleifera* trunk images are shown in Figure 3. For each image, the image name, object classification, and position of the trunk were recorded in .xml file format. All the generated XML files were saved and converted to TXT files.

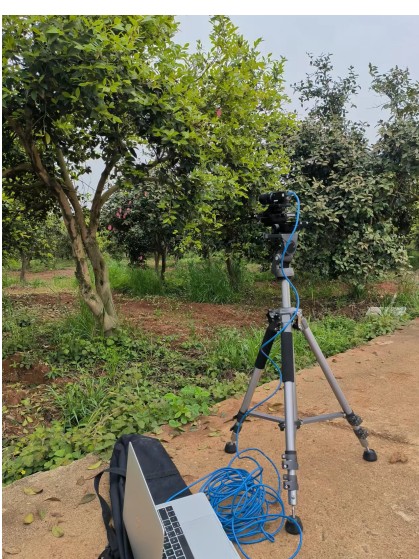

**Figure 1.** View of the images acquistion.

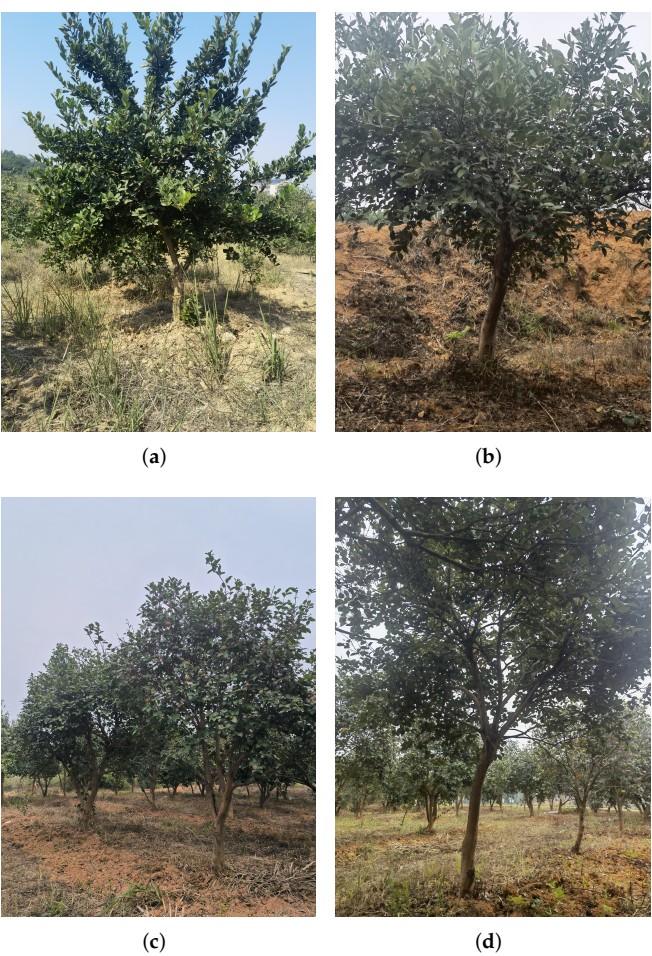

**Figure 2.** Images of *Camellia oleifera* trunk under different conditions. (**a**) Single trunk with front-light. (**b**) Single trunk with back-light. (**c**) Cluster of trunks with front-light. (**d**) Cluster of trunks with back-light.

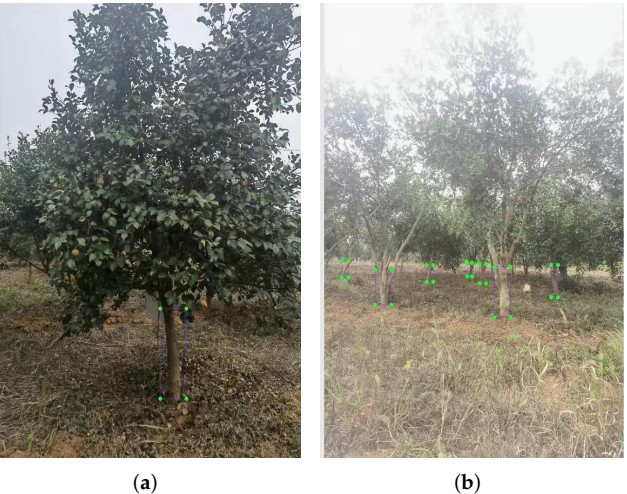

**Figure 3.** Annotation of *Camellia oleifera* trunks in images. (**a**) Single trunk. (**b**) Cluster of trunks.

### 2.3. Data Augmentation

Data augmentation plays a crucial role in deep-learning-based object detection approaches as it enhances data diversity and regularizes the mode [30]. To better extract trunk features and ensure recognition accuracy while avoiding overfitting during training, vari-

ous modification methods were employed for data augmentation. The data augmentation tool and OpenCV were utilized to expand the dataset through operations such as image brightness enhancement, image rotation, image mirror flipping, image random clipping, image noise increases and mosaic. Table 1 provides an explanation of six augmentation operations, and the resulting augmented images are shown in Figure 4. Following the data augmentation process, the *Camellia oleifera* trunk dataset consisted of 9000 (5 × 1500 + 1500) images. From this dataset, 6000 images were randomly selected for the training dataset, 2500 images for test dataset, and 500 images for verification set.

**Table 1.** Data Augmentation applied to the *Camellia oleifera* trunk original images.

| Operation | Value | Description | The Percentage of Total Dataset(%) |
|---|---|---|---|
| Hue, Saturation and Value | Random | Enhance and reduce image's hue, saturation and value | 10 |
| Mirror | Random | Horizontal and vertical mirroring | 10 |
| Noise | Random | Add Gaussian noise | 25 |
| Mosaic | Random | Image Mosaic | 10 |
| Rotation | 90°, 180°, and 270° | Image Rotation | 10 |
| Scale | Random | Image Scale | 20 |

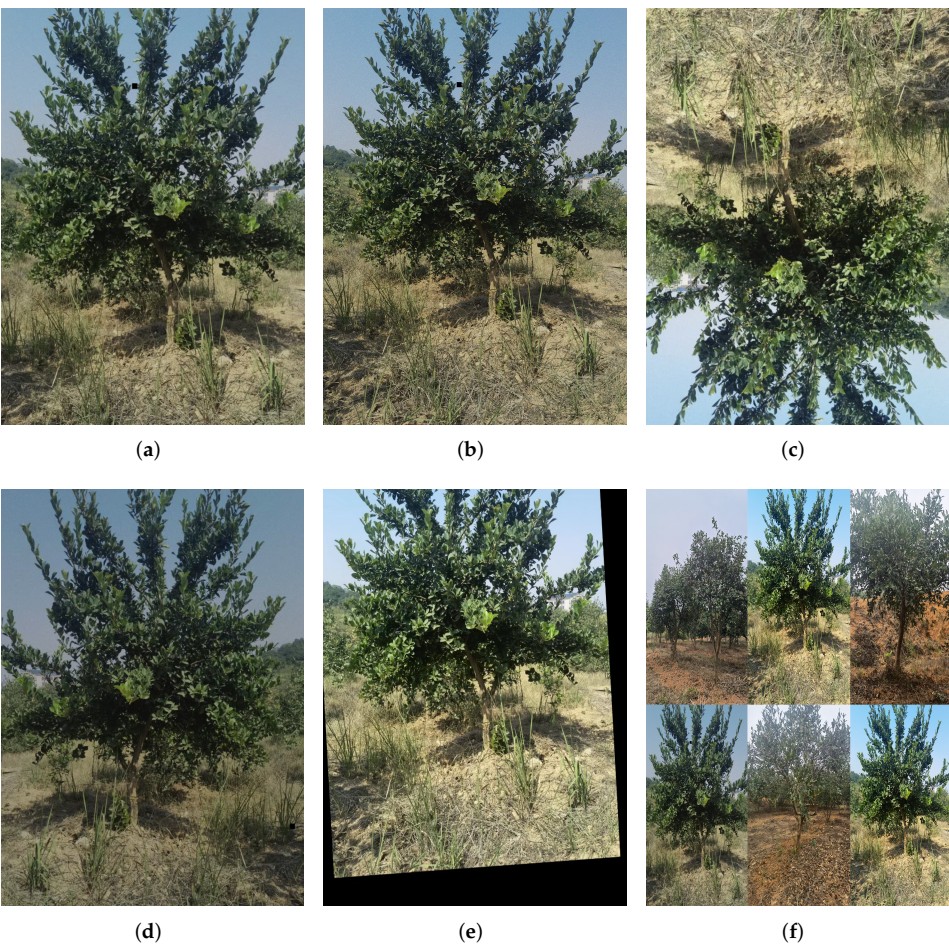

(a)  (b)  (c)

(d)  (e)  (f)

**Figure 4.** Image augmentation results. (**a**) Original image. (**b**) Image noise. (**c**) Image flipping. (**d**) Image mirror. (**e**) Image rotation. (**f**) Image mosaic.

### 2.4. *Camellia oleifera* Trunk Detection Algorithm Based on the Improved YOLOv7

2.4.1. YOLOv7

The current state-of-the-art real-time one-stage object detectors primarily rely on the YOLO series, which is capable of quickly and accurately detecting and classifying targets. This makes it well-suited for meeting the requirements of real-time detection in harvesting robots. In this study, a detection algorithm based on the latest YOLOv7 model is proposed for object detection of *Camellia oleifera* trunks. The YOLOv7 model, proposed by Alexey Bochkovskiy et al. [28], utilizes a re-parameterized approach to replace the original modules and expands the aggregation network. This approach effectively reduces the computational cost of real-time object detection.

The architecture of the YOLOv7 network consists of three main parts, the input network, backbone network and head network, as illustrated in Figure 5. Comparing with YOLOv5, the neck layer and head layer in YOLOv7 are referred to as the head network, and the use of mosaic data augmentation in YOLOv7 is particularly suitable for small object detection [31]. The backbone network is responsible for extracting features, while the head network is used for prediction. Prior to entering the backbone layer, the resolution of each image is preprocessed and resized to 640 × 640 × 3, which is the standard training image size for YOLOv7). Then images are fed into the backbone network. In the backbone layer, the Efficient Layer Aggregation Networks (ELAN) module is employed to replace the CSPDarknet53 network. This module allows the model to learn more features and enhances its robustness. DownC module is utilized for downsampling and the SPPCSP module is redesigned based on CSP [32] to enable richer combinations of gradients and reduce computation. The head network utilizes the outputs from the three layers of the backbone network to generate three different-sized feature maps. Finally, RepVGG blocks and convolutions are used to perform image detection and output the detection results [33].

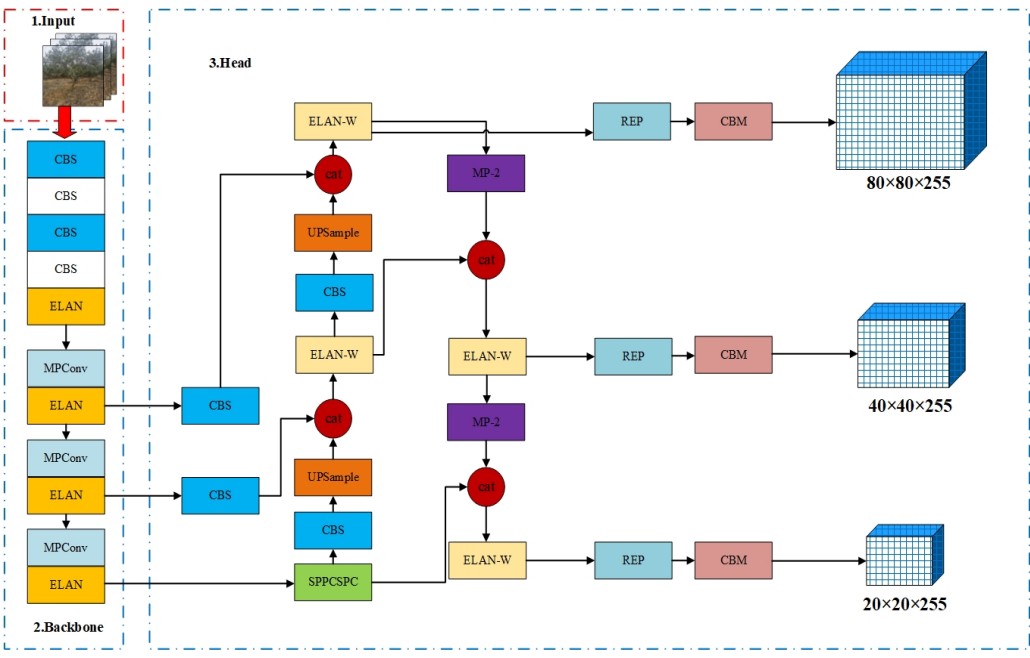

**Figure 5.** The architecture of YOLOv7 network.

2.4.2. Improvement of the YOLOv7 Network

The complex and irregular environment of the *Camellia oleifera* orchard poses challenges for accurate trunk identification. To assist the harvesting robot in detecting and locating the vibration point, it is imperative to improve the detection accuracy by optimizing the network based on YOLOv7.

In order to enhance feature extraction and focus on relevant target objects, attention mechanisms were incorporated into both the backbone and head layers. Attention mecha-

nisms are data processing methods commonly applied in deep learning tasks [34]. They effectively highlight the important and relevant feature information, including channel attention mechanisms [35], spatial attention mechanisms [36], mixed attention mechanisms [37] and so on. These attention mechanisms play a significant role in directing the model's attention to specific regions of interest and improving the overall detection performance.

The CBAM (Convolutional Block Attention Module) block integrates attention maps along two separate channels and spatial dimensions in series for adaptive feature refinement, which emphasizes useful channels as well as enhancing informative local regions [38], as shown in Figure 6. In our study, we applied the CBAM attention model, which incorporates both channel attention model and spatial attention model attention operations. The channel attention model pays more attention to the foreground objects and the meaningful area within the input image, while the spatial attention model focuses on the position information and contextual information across the entire image [39].

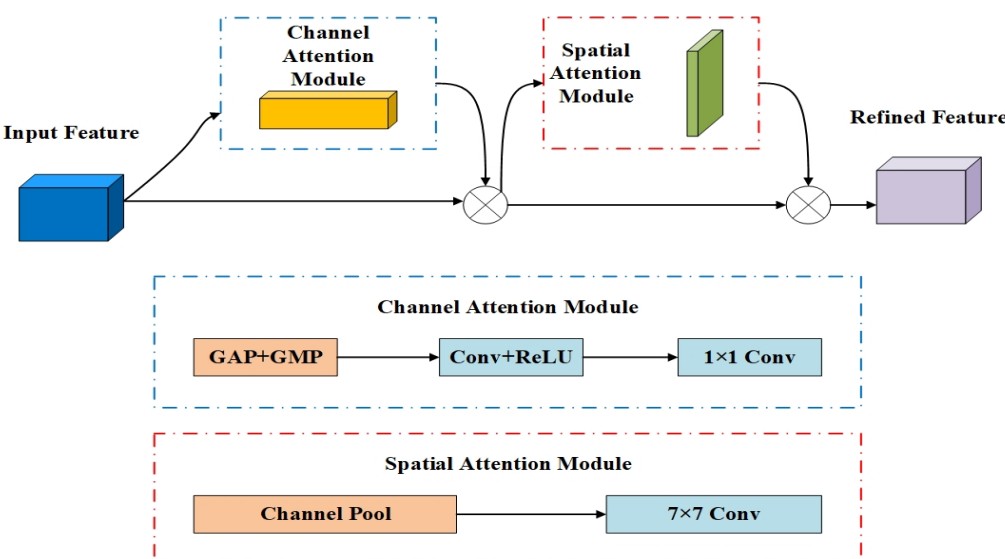

**Figure 6.** The main structure of CBAM attention module.

To preserve the original weights of the backbone network, the CBAM block was added at the beginning of both the backbone and head layers of the YOLOv7 model. The improved YOLOv7 network structure is shown in Figure 7. By incorporating the CBAM model, the feature extraction capabilities of the backbone network are enhanced. This method enables the network to pay more attention to detect key objects and mitigate the impact of interfering elements in complex orchard environments.

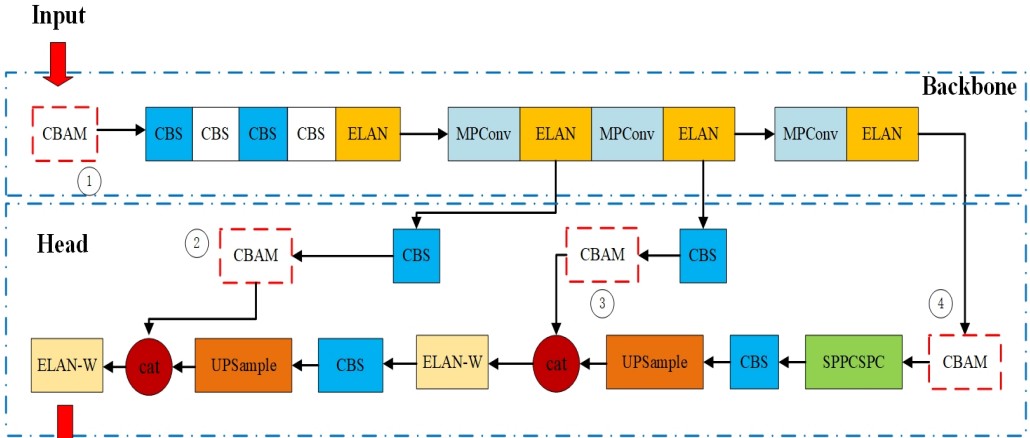

**Figure 7.** The structure of improved YOLOv7.

In the improved backbone network of YOLOv7, the pre-processed image with a size of $640 \times 640 \times 3$ is fed into the backbone. The input feature map undergoes global max pooling (GMP) and global average pooling (GAP) operations, resulting in two feature maps. These two feature maps are then passed through a two-layer multilayer perceptron and activated using the ReLU activation function. The ReLU activation function generates the channel attention feature which is multiplied with the original input feature map to obtain the input feature of the spatial attention module [36]. Finally, the outputs of the channel attention module and the spatial attention module are multiplied together to obtain the output feature map of CBAM. The feature maps from the CBAM are fed into the CBS module in the original backbone, and the final predictions are generated to implement object detection by the model.

### 2.4.3. Loss Function

The loss function of the improved YOLOv7 in our method consists of confidence loss and location loss. The confidence loss is used to gauge the probability that the predicted bounding box contains the actual target and assess the model's performance and identify areas for improvement. Binary cross entropy was employed to calculate the confidence loss in this study; as shown in Equation (1), IoU represents the value of the IoU loss, Ci denotes the prediction confidence, and N refers to the total number of samples.

$$L_{conf} = -\frac{\sum_{i \in N}(IoU \times ln((C'_i))) + (1 - IoU) \times \ln(1 - (C'_i))}{N} \tag{1}$$

$$C'_i = sigmoid(C_i) \tag{2}$$

The location loss is responsible for evaluating the discrepancy between the predicted bounding box and the ground-truth box. In this study, we replaced the original location loss function of YOLOv7 with the Focal-EIoU loss [40]. Focal-EIoU loss is the combination of focal loss and EIoU loss. The focal loss [41] dynamically scales the cross-entropy loss to address the significant class imbalance between foreground and background classes during training, thereby enabling high-accuracy detection in one-stage object detection scenarios. The EIoU loss function directly measures the overlap area, central point and side length of targets, and anchors to ensure convergence speed and localization accuracy [39]. The Focal-EIoU loss, as shown in Equation (3), incorporates both the focal loss and the EIoU loss. Here $b$ and $b^{gt}$ represent the central points of the predicted box and ground-truth box, while $w$ and $h$ are the length and width of ground-truth box, and $w^{gt}$ and $h^{gt}$ represent the length and width of the predicted bounding box. Furthermore, $\rho$ is the Euclidean distance between the predicted box and the ground-truth box, $w^c$ and $h^c$ donate the width and height of the smallest enclosing box that covers the two boxes, and $W_i$ is a parameter that controls the degree of suppression of outliers. Facol-EIoU is designed to address the limitations of CIoU by minimizing the difference between the width and height of the target box and the anchor, resulting in faster convergence and better localization results.

$$L_{EIoU} = 1 - IoU + \frac{\rho^2(b, b^{gt})}{(w^c)^2 + (h^c)^2} + \frac{\rho^2(w, w^{gt})}{(w^c)^2} + \frac{\rho^2(h, h^{gt})}{(h^c)^2} \tag{3}$$

$$L_{loc} = \frac{\sum_{i=1}^{n} W_i \times L_{EIoU}}{\sum_{i=1}^{n} W_i} \tag{4}$$

Combined with location loss and confidence loss, the loss function of the improved YOLOv7 in this study is shown in Equation (5).

$$Loss = L_{conf} + L_{loc} \tag{5}$$

### 2.5. Model Training

2.5.1. Training Platforms and Parameter Settings

In this study, the target detection model was implemented using the PyTorch deep learning framework. The experiments were conducted on a system equipped with an Intel Core i7 CPU and an NVIDIA GeForce RTX2070 GPU with 12 GB of video memory, running on the Windows 10 operating system. Python 3.8 served as the programming language, and the training and testing of the model were carried out using CUDA 11.3, CUDNN 8.2, OpenCV 3.4.5 and Pycharm2016. The training configuration consisted of using the stochastic gradient descent (SGD) optimizer with an initial learning rate of $1 \times 10^{-2}$, a momentum of 0.937, and a weight decay of 0.0005.

The batch size was set to 16 and training was 600 epochs. The input size of images was set to $640 \times 640$ pixels. The momentum decay and weight decay were both set to 0.937 and 0.0005. During training, the initial vector was set to 0.01 and the IOU threshold was set to 0.5. Data augmentation was applied with the coefficient of hue (H), saturation (S) and lightness (V) set to 0.2, 0.5 and 0.4. Throughout the training process, the training data, loss values and model weights were saved at each epoch. The performance of the model was evaluated using the test set. Six detection algorithms were trained and compared in this study: YOLOv3, YOLOv4, YOLOv5, YOLOv7 and the improved YOLOv7. The training process used the Adam optimizer with 600 training epochs. To prevent overfitting, the training would automatically stop if there was no improvement in accuracy over the last 50 training epochs. The training parameters used in the experiments are summarized in Table 2.

**Table 2.** Training Parameter.

| Parameter | Value | Parameter | Value |
|---|---|---|---|
| Ephochs | 600 | Batch Size | 16 |
| Learning Rate | 0.01 | Weight Decay | 0.0005 |
| Image Size | $640 \times 640$ | Momentum | 0.937 |

2.5.2. Evaluation Indicators of the Model

In this paper, the performance of the detection model was evaluated using Precision (*P*), Recall (*R*), Mean Average Precision (*mAP*) and *F1* [41] score. A higher mAP score indicates better performance. The mAP50 represents the mAP at an IoU threshold of 0.5, while mAP75 refers to the mAP at an IoU threshold of 0.75. The overall mAP is calculated as the average of mAP values across IoU thresholds ranging from 0.5 to 0.95, with an interval of 0.05. The evaluation metrics rely on the computation of true positives (*TP*), false positives (*FP*), and false negatives (*FN*), as shown in Equation (6). Precision measures the ratio of correctly detected targets to the total number of detected targets and serves as an intuitive index for detection evaluation. The (*mAP*) Recall and F1 score were introduced to provide a comprehensive evaluation.

$$P = \frac{TP}{TP + FP} \tag{6}$$

$$R = \frac{TP}{TP + FN} \tag{7}$$

$$AP = \int_0^1 P(R)dR \tag{8}$$

$$mAP = \frac{1}{n} \sum_{i=1}^{n} AP_i \tag{9}$$

$$F1 = 2 \times \frac{P \times R}{P + R} \tag{10}$$

In this study, the detection task focused on a single category, which is the *Camellia oleifera* trunk. Therefore, the *AP* and *mAP* values are equal since there is only one category involved in the detection process. *TP* is the number of *Camellia oleifera* trunks that are detected correctly. *FP* is the number of negative objects that were detected as the *Camellia oleifera* trunk, and *FN* is the number of positive samples incorrectly detected.

## 3. Results

In order to evaluate the performance of our model in *Camellia oleifera* trunk detection, this section provides a detailed description of the experiments conducted, including the training process, experimental results and ablation experimental results. A comparison is made between different detection algorithms to assess the contributions of the methods proposed in this paper. The test set consists of 2500 images, categorized as follows: 600 images of single trunks with front-light, 600 images of single trunks with back-light, 600 images of clusters of trunks with front-light, and 700 images of clusters of trunks with back-light. Our model is compared with other networks such as YOLOv3, YOLOv4, YOLOv5 and the original YOLOv7, using the images from the test set. The workflow of the proposed study is shown in Figure 8.

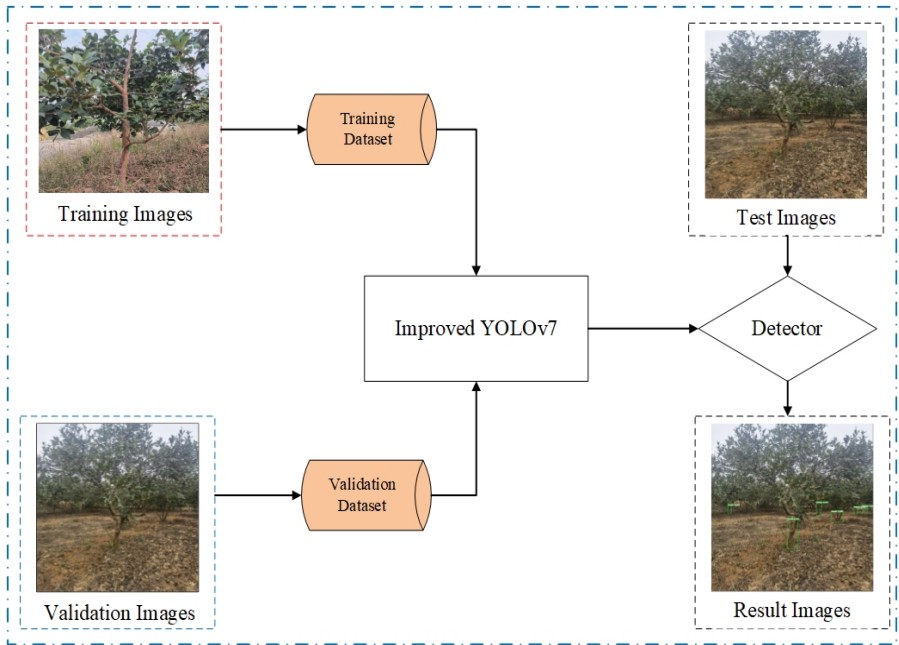

**Figure 8.** Workflow of the proposed study.

### 3.1. Training Results

The loss curves of the training and validation during the training process are shown in Figure 9. The curves demonstrate a rapid decrease in loss value during the initial 150 epochs, followed by a steady decline beyond 600 epochs. Notably, there was no evidence of overfitting, and the results were satisfactory. Based on these observations, the model trained for 600 epochs was deemed suitable for detecting *Camellia oleifera* trunks, thus serving the purpose of the *Camellia oleifera* fruit harvesting robot.

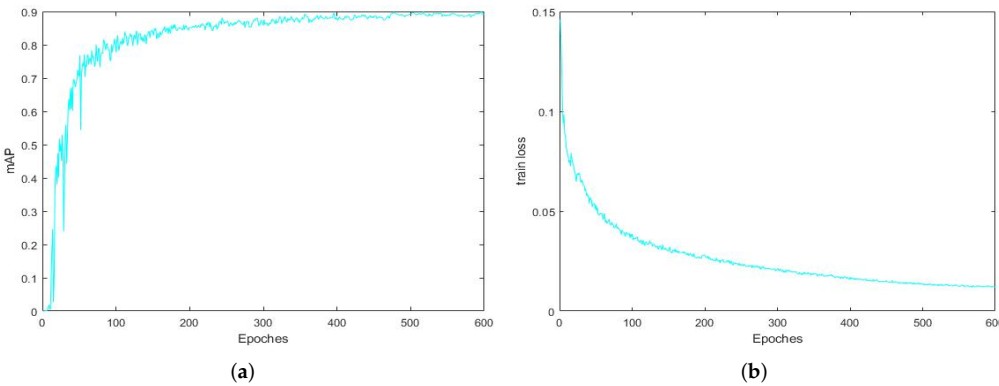

**Figure 9.** Training Results. (**a**) mAP curve. (**b**) Loss curve.

### 3.2. Ablation Experiments

Ablation experiments were conducted to assess the impact of the CBAM and Focal-EIoU loss modules on the detection performance. Table 3 presents the results of different combinations and their respective detection metrics. Four models were compared: YOLOv7 without using CBAM and Focal-EIoU loss function, YOLOv7 improved with CBAM only, YOLOv7 improved with Focal-EIoU loss function only, and YOLOv7 improved with CBAM and Focal-EIoU loss function. The performance metrics of these models were evaluated and compared.

**Table 3.** Results of ablation experiments.

| CBAM | Focal-EIoU Loss Function | mAP(%) | Recall | F1 |
|:---:|:---:|:---:|:---:|:---:|
|  |  | 84.2 | 0.89 | 0.84 |
| √ |  | 86 | 0.92 | 0.84 |
|  | √ | 85.6 | 0.9 | 0.84 |
| √ | √ | 89.2 | 0.94 | 0.87 |

Results in Table 3 demonstrate that the accuracy of the models varies depending on the combination of the CBAM and Focal-EIoU loss function. In terms of mAP metric, the introduction of CBAM improves the YOLOv7 model by 2.1%. Additionally, the introduction of the Focal-EIoU loss function improves the YOLOv7 model by 1.7% , which means the CBAM enhances the model to focus on detection targets and the Focal-EIoU loss function improves the accuracy of the detection. When both CBAM and Focal-EIoU loss function are applied in YOLOv7, The mAP of model improves by 5.9% and the detection results achieve the optimal detection performance.

### 3.3. Experiment Results

To verify the effectiveness of the improved detection network model for *Camellia oleifera* trunks, test datasets consisting of trunk images were applied in the experiments. The detection results for single trunks with front-light, single trunks with back-light, clusters of trunks with front-light and clusters of trunks with back-light were analyzed. As shown in Figure 10, the majority of trunks in the orchard were identified even in the presence of occlusions and different light conditions. The detection results obtained from our model were found to be excellent.

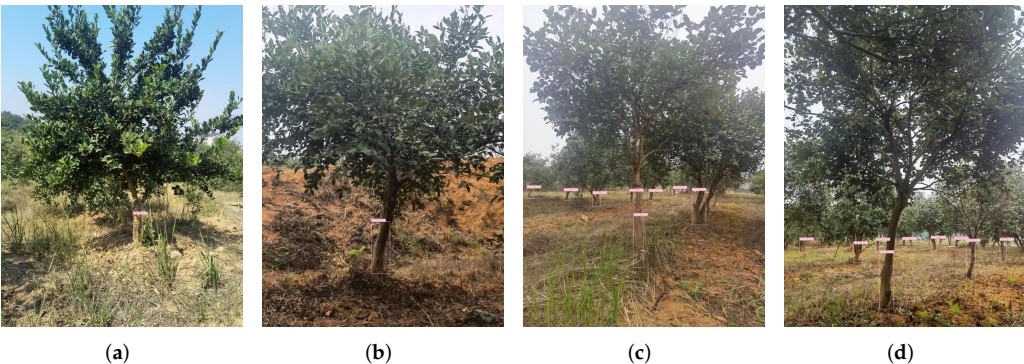

|  (a)  |  (b)  |  (c)  |  (d)  |

**Figure 10.** Experimental results of trunk detection under different conditions. (**a**) Single trunk with front-light. (**b**) Single trunk with back-light. (**c**) Cluster of trunks with front-light. (**d**) Cluster of trunks with back-light.

*3.4. Comparison of Detection Algorithms*

To demonstrate the recognition performance of the improved YOLOv7 network for *Camellia oleifera* trunks, we compared models proposed in our study with the YOLOv3, YOLOv4, YOLOv5 and YOLOv7 networks using the test dataset. The metrics used for comparison were mAP, Recall, F1 and Average Detection Speed. All object detection algorithms were trained using the same dataset as described in this study.

We compared the effectiveness of detecting *Camellia oleifera* trunks with different models, and the results are presented in Table 4 and Figure 11. It can be observed that our model achieves the highest mAP, which is 1.05 times higher than YOLOv3, 1.1 times higher than YOLOv4, 1.06 times higher than YOLOv5 and 1.05 times higher than YOLOv7. The Average Detection Speed of our model is 0.018 s/pic, which is 0.5 times, 0.4 times, 0.43 times and 0.72 times faster than those of the YOLOv3, YOLOv4, YOLOv5 and YOLOv7 networks. These results indicate our model performed better in detecting *Camellia oleifera* trunks and could meet the requirements of *Camellia oleifera* fruit harvesting robots for trunk recognition.

**Table 4.** Comparison of different models of *Camellia oleifera* trunks detection.

| Object Detection Networks | mAP (%) | Recall | F1 | Average Detection Speed (s/pic) |
|:---:|:---:|:---:|:---:|:---:|
| YOLOv3 | 84.9 | 0.83 | 0.84 | 0.032 |
| YOLOv4 | 80.9 | 0.82 | 0.81 | 0.045 |
| YOLOv5 | 83.9 | 0.88 | 0.83 | 0.041 |
| YOLOv7 | 84.2 | 0.89 | 0.84 | 0.025 |
| Our model | 89.2 | 0.94 | 0.87 | 0.018 |

The comparison results demonstrate that our model not only ensures high detection accuracy but also meets the detection speed requirements of the harvesting robot. The improved YOLOv7 proposed in this study achieves excellent recognition results under different light conditions. Our model exhibits faster detection speed compared to YOLOv3, YOLOv4, YOLOv5 and YOLOv7. As shown in Figures 12 and 13, our model is capable of recognizing more trunks that are out of focus in the camera, indicating its robustness. Particularly, under back-light conditions, YOLOv3, YOLOv4 and YOLOv5 exhibit insensitivity in detecting small objects, resulting in the failure to detect small trunks on the left side of the image (as shown in Figure 13b,d,f,h,j). This is likely due to the lack of sufficient color features of *Camellia oleifera* trunks in back-light conditions. The improved YOLOv7 and Focal-EIoU loss function generate better trunk features, effectively enhancing the performance and detection speed of our model.

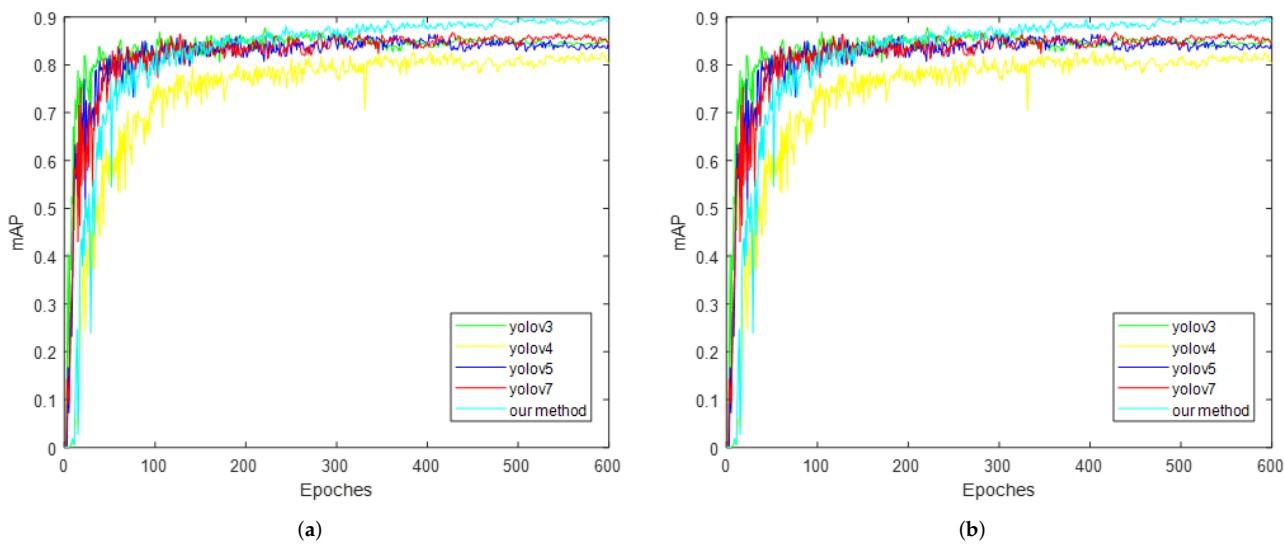

(**a**)　　　　　　　　　　　　　　　　　(**b**)

**Figure 11.** Comparison of different detection models. (**a**) mAP curve. (**b**) Loss curve.

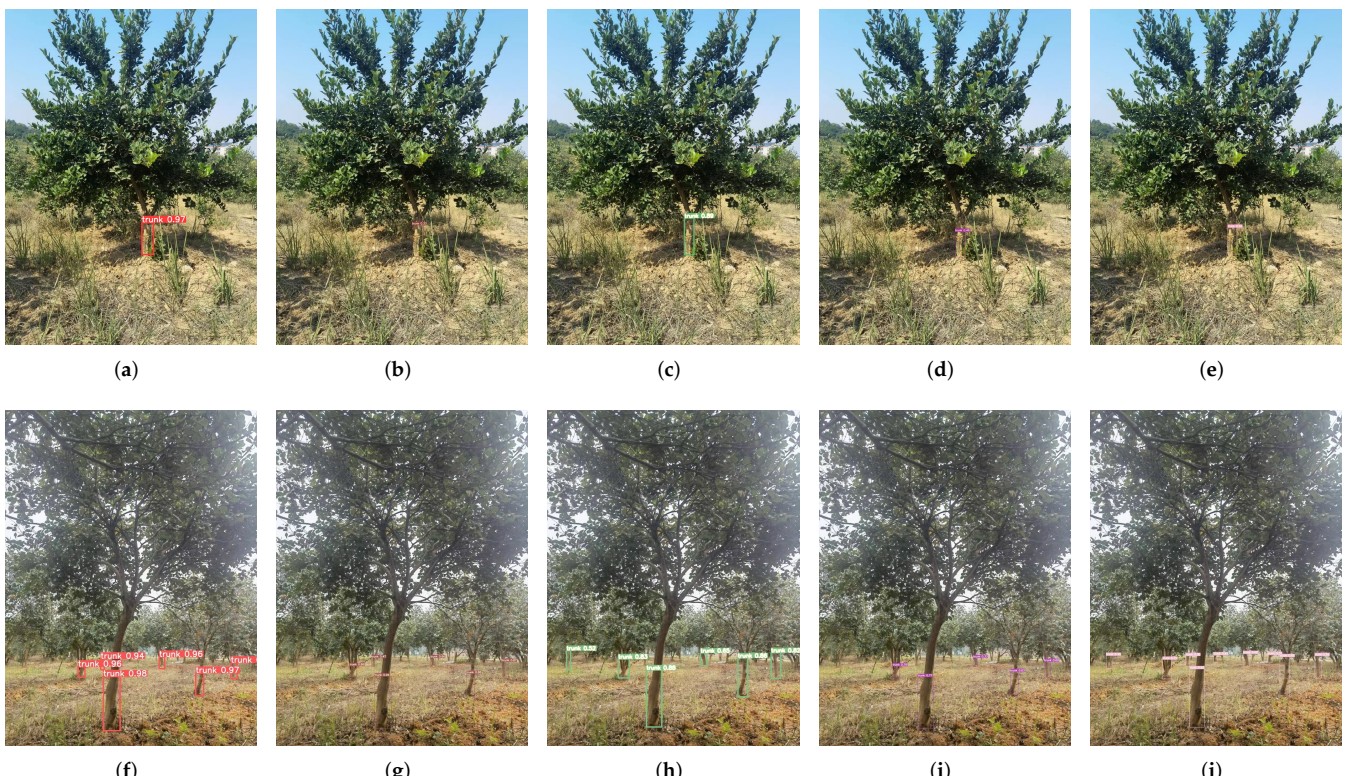

**Figure 12.** Comparison results of *Camellia oleifera* trunk detection with front-light by different models. (**a**)YOLOv3 (Single trunk). (**b**) YOLOv4 (Single trunk). (**c**) YOLOv5 (Single trunk). (**d**) YOLOv7 (Single trunk). (**e**) Our model (Single trunk). (**f**) YOLOv3 (Cluster of trunks). (**g**) YOLOv4 (Cluster of trunks). (**h**) YOLOv5 (Cluster of trunks). (**i**) YOLOv7 (Cluster of trunks). (**j**) Our model (Cluster of trunks).

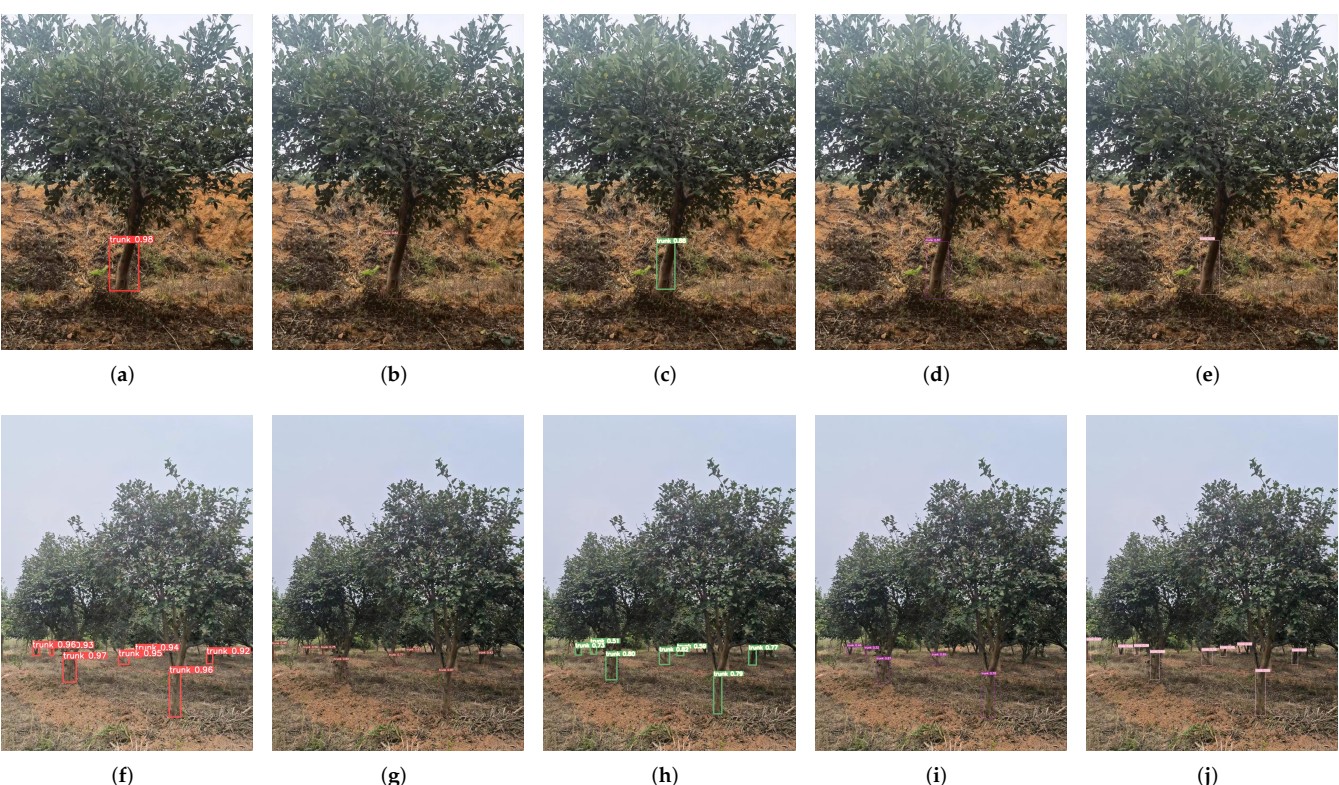

**Figure 13.** Comparison results of *Camellia oleifera* trunks detection with back-light by different models. (**a**) YOLOv3 (Single trunk). (**b**)YOLOv4 (Single trunk). (**c**) YOLOv5 (Single trunk). (**d**) YOLOv7 (Single trunk). (**e**) Our model (Single trunk). (**f**) YOLOv3 (Cluster of trunks). (**g**) YOLOv4 (Cluster of trunks). (**h**) YOLOv5 (Cluster of trunks). (**i**) YOLOv7 (Cluster of trunks). (**j**) Our model (Cluster of trunks).

## 4. Discussion

In a forest crop harvesting application, trunk detection can help the harvesting robot to locate the trunk. However, due to the complex environment of orchards, the accuracy of tree trunk detection is affected by different lighting and shadows. Our research contributes to trunk detection with a comparison of different detection methods. Su, F et al. [14] proposed a tree trunk and obstacle detection method in a semi-structured apple orchard environment based on improved YOLOv5s, without considering different light conditions. Yang et al. [42] proposed a method of Y3TM to detect trunks, streetlight and telephone poles in the forest. The presented research demonstrates that the deep-learning-based detection algorithm has excellent performance in an unconstructed environment and provides the basis for harvesting robots.

Compared with relevant studies, the *Camellia oleifera* trunk detection method based on an improved YOLOv7 proposed in this paper has better recognition accuracy and faster speed. The application of the attention mechanism CBAM in the YOLOv7 backbone and the replacement of the original loss function with the Focal-EIoU loss function have significantly contributed to enhancing the intelligence of the *Camellia oleifera* fruit harvesting robot, enabling it to determine vibration points using the vision sensing system. The training results demonstrate that the improved YOLOv7 model can effectively prevent overfitting and achieve faster convergence during the training process. The ablation experiment results indicate that our model outperforms the other recognition methods in terms of both detection speed and accuracy for *Camellia oleifera* trunk recognition. The results of comparison experiments show that our model can achieve the best detection results over the mainstream detection algorithms under different light conditions. In future research, the improved YOLOv7 model can also applied to detect *Camellia oleifera* fruit,

*Camellia oleifera* flowers and other types of forest crops. Thus, the improved YOLOv7 model presented in this paper has the potential to not only contribute to the fields of *Camellia oleifera* trunks detection, but also advanced crop detection in smart forestry applications.

## 5. Conclusions

The detection and recognition of *Camellia oleifera* trunks in unstructured environments is a crucial technology for the advancement of *Camellia oleifera* fruit harvesting robots. In this paper, we propose a highly accurate and efficient method for detecting and recognizing*Camellia oleifera* trunks. Our approach utilizes an improved YOLOv7 network as the detection model. By integrating the CBAM attention mechanism into the YOLOv7 backbone, we enhance the model's detection accuracy. Additionally, we adopt the Facol-EIoU loss function, which replaces the original YOLOv7 location loss function, improving the model's robustness. Various well-known detection networks are compared and analyzed for their performance in *Camellia oleifera* trunk detection. Experimental results demonstrate that our proposed detection model outperforms other methods in terms of accuracy and speed when applied to the *Camellia oleifera* trunk dataset. The development of intelligent *Camellia oleifera* fruit harvesting robots holds profound significance for the *Camellia oleifera* industry. This study offers an effective vision system solution for *Camellia oleifera* fruit harvesting robots. In the future, we will continue to advance object detection technology in harvesting robots and update the detection method to enable intelligent harvesting.

**Author Contributions:** Y.L. (Yang Liu) and L.L. proposed the trunk detection method and designed the research; H.W. and Y.L. (Yang Liu) performed the experiments; H.W., Y.L. (Yuanyin Luo), H.L., Y.L. (Yinhui Liu), H.C. and K.L. collected and analyzed the experiment data; Y.L. (Yang Liu) and H.W. wrote the manuscript. All authors have read and agreed to the published version of the manuscript.

**Funding:** This research was supported by National Key Research and Development Program (2022YFD2202103), Scientific Innovation Fund for Post-graduates of Central South University of Forestry and Technology (2023CX01025).

**Data Availability Statement:** The data presented in this study are available on request from the corresponding author.

**Conflicts of Interest:** The authors declare that there are no conflict of interest.

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
