# Peer review of "A Trunk Detection Method for Camellia oleifera Fruit Harvesting Robot Based on Improved YOLOv7"

_forests, doi:10.3390/f14071453_

Round 1

Reviewer 1 Report

The paper titled : A Trunk Detection Method for Camellia oleifera Fruit Harvesting Robot Based on improved YOLOv7 aimed to enhance trunk detection performance in unstructured environments using improved YOLOv7.
I consider the topic original and relevant in the field
The innovation is to develop a robust and reliable approach to detect Camellia oleifera trunks using an improved YOLOv7 model.
The methdology is clear and simple to understand
The results and discussion must be simplified and highlight the add value of the subject and the developped model
The conclusions is consistent with the evidence and arguments presented. However it is important to add the importance of the developed tool and its application in the field.

Reviewer 2 Report

The study is of the nature to make important contributions to the scientific field. The improved YOLOv7 model exhibits excellent trunk detection accuracy, enabling Camellia oleifera harvesting robots to effectively detect trunks in unstructured orchards. However, some corrections must be made.

References to previous work written on line 29 should be cited.

The explanations of the abbreviations on line 50 should also be written.

Figure numbers should be written on line 329.

Discussion section could be improved.

The possible agricultural advantages of this proposed method should also be mentioned in the conclusion.

Spelling errors in references should be corrected. Scientific species names should be written in italics.

Sentences similar to the article below should be replaced: Detection of Camellia oleifera Fruit in Complex Scenes by Using YOLOv7 and Data Augmentation

Reviewer 3 Report

I find this an interesting and well written article. Please find below some suggestions.

please revise the keywords, they shouldn't repeat the words of the title

LINE 23 Camellia oleiferafruit please add a space

Introduction too long please revise it.

Please remove the last part of the Introduction: "The rest of the paper is organized as follows: Section 2 describes the process of 94 image acquisition, data augmentation and the improved trunk detection model; Section 3 95 presents the training configurations and evaluation format used to assess the models. The 96 results of this work, including ablation experiment results and a comparison of different 97 deep learning detection algorithms, are are presented in section 3. This is followed by a 98discussion and conclusions in Section 4 and Section 5, respectively. The main findings of 99 this study are described, and suggestions for future work are proposed."

the quality of figure 11 is very low.

line 329 and 333 "Figure??" please insert the number

the discussion has to be expanded, is too small, and a comparison citing other papers from the literature should be done.

please add in the conclusions section some details about the harvesting robots.

Good luck for you manuscript

Round 2

Reviewer 3 Report

I would like to sincerely thank the authors for the improvements they made according to my suggestions. I suggest publication in its present form.